# Genome-Wide Analysis of the BAHD Family in Welsh Onion and CER2-LIKEs Involved in Wax Metabolism

**DOI:** 10.3390/genes14061286

**Published:** 2023-06-18

**Authors:** Lecheng Liu, Huanhuan Xu, Wanyue Zhang, Jiayi Xing, Mingzhao Zhu, Yuchen Zhang, Yongqin Wang

**Affiliations:** 1College of Horticulture and Gardening, Yangtze University, Jingzhou 434025, China; lchliu18@yangtzeu.edu.cn (L.L.);; 2Beijing Vegetable Research Center, Beijing Academy of Agricultural and Forestry Sciences, Key Laboratory of Biology and Genetic Improvement of Horticultural Crops (North China), Ministry of Agriculture, Beijing Key Laboratory of Vegetable Germplasm Improvement, National Engineering Research Center for Vegetables, Beijing 100097, China; 3Department of Horticulture, Agricultural College, Shihezi University, Shihezi 832003, China

**Keywords:** BAHD acyltransferase, CER2-LIKEs, genes family, abiotic stress, epicuticular wax

## Abstract

BAHD acyltransferases (BAHDs), especially those present in plant epidermal wax metabolism, are crucial for environmental adaptation. Epidermal waxes primarily comprise very-long-chain fatty acids (VLCFAs) and their derivatives, serving as significant components of aboveground plant organs. These waxes play an essential role in resisting biotic and abiotic stresses. In this study, we identified the BAHD family in Welsh onion (*Allium fistulosum*). Our analysis revealed the presence of AfBAHDs in all chromosomes, with a distinct concentration in Chr3. Furthermore, the *cis*-acting elements of AfBAHDs were associated with abiotic/biotic stress, hormones, and light. The motif of Welsh onion BAHDs indicated the presence of a specific BAHDs motif. We also established the phylogenetic relationships of AfBAHDs, identifying three homologous genes of CER2. Subsequently, we characterized the expression of AfCER2-LIKEs in a Welsh onion mutant deficient in wax and found that AfCER2-LIKE1 plays a critical role in leaf wax metabolism, while all AfCER2-LIKEs respond to abiotic stress. Our findings provide new insights into the BAHD family and lay a foundation for future studies on the regulation of wax metabolism in Welsh onion.

## 1. Introduction

The BAHDs family of acyltransferases is one of the largest metabolic protein families in land plants [1]. These enzymes play a crucial role in plant metabolism and are involved in the acetylation of various plant metabolites. Despite consisting of many acyltransferases, the BAHD family derives its name from the first letter of four types of enzymes within the family. BAHD members are essential during the transition of plants from aquatic to terrestrial environments and play a key role in both abiotic and biotic stress responses [2,3,4]. BAHD acyltransferases are known for their broad substrate specificity and can be divided into nine classes on the basis of the specificity of their substrates, further illustrating the importance of BAHDs in plants. Additionally, their diverse substrate specificity makes BAHDs useful in synthetic biology, where BAHD genes have been introduced into microorganisms to produce target molecules. Most BAHDs contain conserved HXXXD and DFGWG motifs, despite exhibiting low similarity and distinct functions [5]. Multiple studies have identified BAHD gene members in various plants, with 55 members in *Arabidopsis*, 100 members in *Populus*, and 84 members in rice [6,7]. These studies have also supported the role of BAHDs in plant growth, development, and resistance to biotic/abiotic stresses through the analysis of BAHD gene expression levels. Therefore, understanding the BAHD genes in different plant genomes holds significant importance. BAHDs members involved in various metabolisms to ensure plant adaptation to environmental stress [8,9,10,11].

The first reported BAHD gene in plants was *glossy2*/*CER2* (ECERIFERUM2), and mutations of this gene were found to cause wax deficiency in maize, rice, and *Arabidopsis thaliana* [12,13,14,15]. Epicuticular wax serves as a natural barrier for plants and plays a crucial role in resistance to both biotic and abiotic stress, especially in drought stress [16,17,18]. Furthermore, plants can adapt to the environment by producing derivatives modified by BAHDs. For example, the expression of *PtFHT* (feruloyl transferase) from poplar in *Arabidopsis* can effectively enhance their salt tolerance by increasing ferulate modifications. Other members of the BAHD family are involved in suberin biosynthesis, such as ASFT (aliphatic suberin feruloyl transferase), HFT (hydroxycinnamoyl transferase), and DCR (defective of cuticular ridges) [19,20,21,22]. Mutation of the AtASFT function results in significantly reduced levels of ferulic acid monomers in seed coat suberin. The accumulation of suberin in plants can alleviate environmental stress such as drought and salinity. BAHDs are also associated with lignin metabolism. Silencing the HST (hydroxycinnamoyl-CoA: shikimate hydroxycinnamoyl transferase) gene in *Arabidopsis* and *Nicotiana benthamiana* can change lignin composition and result in a dwarf phenotype [23]. Additionally, BAHDs are necessary for pollen wall generation, with AtSHT (hydroxycinnamoyl-CoA: spermidine hydroxycinnamoyl transferase) being related to cinnamoyl spermidine derivatives, and mutations of SHT leading to irregularities in the pollen wall [24,25,26]. BAHDs are also involved in the acylation of flavonoids, and elevated levels of these metabolites in flower parts of plants contribute to UV protection [27,28]. Furthermore, anthocyanins can serve as substrates for acyltransferases, and acetylated anthocyanins exhibit higher stability, which is beneficial for plants in adapting to UV-B exposure. In summary, recent studies have demonstrated the diverse functions of BAHDs in plants and their significance in plant development and response to various stresses.

Welsh onion is a biennial herb widely cultivated in China and Southeast Asia. It is rich in bioactive substances, including flavonoids, carbohydrates, and sulfur compounds, which possess bactericidal and anti-inflammatory effects. As a result, Welsh onion has been used in traditional medicine for disease prevention [29,30]. The leaf epidermis of Welsh onion is noticeably covered in wax [31]. Homologs of CER2 within the BAHD family play a crucial role in the metabolism of very-long-chain fatty acids (VLCFAs) [32]. However, there have been no reports on the presence of BAHDs in Welsh onion, and the role of CER2-LIKEs in Welsh onion remains unclear.

In this study, we performed a comprehensive investigation of the BAHDs gene family in Welsh onion through genome-wide analysis and identified three homologous genes associated with CER2. We further characterized the key features of these genes through bioinformatics analysis. Additionally, we analyzed the expression patterns of these identified genes under abiotic stress conditions. The results suggest that *AfCER2-LIKEs* are involved in cold stress, heat stress, salt stress, and drought stress. Overall, our study aims to lay the foundation for further research on the metabolism of very-long-chain fatty acids in Welsh onion.

## 2. Materials and Methods

### 2.1. Plant Materials and Treatments

In this study, we utilized the ‘BianGan’ variety of Welsh onion (BG) supplied by the Beijing Vegetable Research Center. Additionally, the waxy mutant of Welsh onion, known as GLBG, was used, which was reported previously by Liu and exhibits significant differences in cuticular wax compared to BianGan. For gene expression analyses in different tissue samples, flowering Welsh onion was used as the plant material. For gene expression analyses under different abiotic stress conditions, we used 2 month old Welsh onion plants grown in a greenhouse under natural light at 27 °C. The seedlings were incubated at 10 °C for cold stress exposure and at 35 °C for heat stress exposure. To apply drought stress, the seedlings were carefully transplanted into dry soil. Lastly, for salt stress, the 2 month old seedlings were treated with a 1% NaCl solution. The Welsh onion samples were harvested 0 h, 4 h, 12 h, 24 h, 48 h, and 72 h after respective stress treatments and stored at −80 °C.

### 2.2. Identification of the AfBAHDs Family in Welsh Onion Genome

We acquired the genome sequence of Welsh onion from CNSA (https://db.cngb.org/cnsa/, accessed on 3 May 2022) under the accession CNP0002276 for subsequent identification and analysis. The identification of BAHD gene family members was performed using the hidden Markov model (HMM) method with an *E*-value threshold of 0.001 on TB tools. The HMM of PF02458 was downloaded from Pfam (http://pfam-legacy.xfam.org/, accessed on 3 May 2022). After conducting a search using the HMM of PF02458, a new HMM file was generated on the basis of the results, and a further search was conducted in the Welsh onion database. For the characterization of conserved domains in the identified AfBAHDs, the CD-search tools were used. Through domain screening, genes containing conserved BAHD domains were retained for subsequent analysis. The MEME software was used to analyze the motifs of AfBAHDs, resulting in the identification of 10 conservative motifs, which were further examined.

### 2.3. Phylogenetic, Protein Properties, and Sequence Analyses

For phylogenetic analyses, we compared the BAHDs reported in previous studies with the identified AfBAHDs using MEGA6. The following BAHDs were included in the analysis: AtDCR (At5g23940), LuDCR (AHA57444), AtCER26 (NP_193120.1), AtCER2 (NP_194182.1), ZmCER2 (CAA61258.1), HCBT (O23917.1), HQT (NP_001312079.1), AsHHT (Q7XXP3.1), AtHCT (NP_199704.1), NtHCT (NP_001312552.1), ZmPCAT (NP_001149738.2), OsAT4 (NP_001403263.1), OsAT10 (XP_015641801.1), AtASFT (NP_851111.1), StFHT (NP_001275190.1), AtFACT (NP_201161.1), and AtDCF (NP_190441.1). The phylogenetic tree was generated using the neighbor-joining method with 1000 bootstrap replicates. The p-distance method and pairwise deletion method were used to resolve gaps in the amino-acid sequences. The protein properties of the AfBAHDs were predicted using ExPASy tools (http://expasy.org/, accessed on 3 May 2022), and the molecular weight and theoretical isoelectric point data of all AfBAHDs were obtained using this online tool. The gene structures of AfBAHDs were analyzed using the GSDS tool (http://gsds.cbi.pku.edu.cn/, accessed on 3 May 2022). Three-dimensional (3D) modeling of AfCER2-LIKE proteins was performed using the Phyre2 online tools with default settings (http://www.sbg.bio.ic.ac.uk/phyre2/html/page.cgi?id=index, accessed on 3 May 2022).

### 2.4. Chromosome Location of the BAHDs in Welsh Onion

To analyze the chromosome locations of AfBAHDs, we used TBtools to parse the GFF3 gene positions and acquire the chromosome annotation for each AfBAHD. We then generated a chromosome map of the genes using Mapchart software. Each Mb base pair within a chromosome was considered a unit. All identified AfBAHDs were labeled on their respective locations within each chromosome map.

### 2.5. cis-Acting Element Analysis of the BAHDs Genes in Welsh Onion

To analyze the *cis*-acting elements, we retrieved the promoter sequences of AfBAHDs from the Welsh onion genome. Specifically, we obtained the sequences located 2000 bp upstream of the *AfBAHDs*. These promoter sequences were then submitted to PlantCARE online tools for analysis. Subsequently, the results were filtered according to the annotations of the *cis*-acting elements.

### 2.6. RNA Extraction and RT-qPCR Analysis

The total RNA of Welsh onion was extracted as previously described by Liu [33], and cDNA synthesis was performed according to the protocol provided by the Hiscript III First-Strand cDNA Synthesis Kit (Vazyme, Nanjing, China). Quantitative PCR (qPCR) was performed on a LightCycler 480 as described previously [31] with the following steps: 2 min at 98 °C followed by 45 cycles of 30 s at 95 °C, 30 s at 58 °C, and 60 s at 72 °C. The coding sequences of AfCER2-LIKE1 (AfisC6G05257), AfCER2-LIKE2 (AfisC8G02662), and AfCER2-LIKE3 (AfisC5G00688) were acquired from the Welsh onion genome database. The primers for qPCR were designed using real-time PCR online tools of Integrated DNA Technologies and obtained from Tsingke Biotechnology Co., Ltd, Beijing, China. Each experiment was performed with three independent biological replicates, and the experimental results were analyzed by the 2^−∆∆Ct^ method.

### 2.7. RNA-Seq and Differential Expression Genes Analysis

The RNA-seq dataset was generated from our previous study [29] and performed by Beijing Biomarker Technologies using the Illumina HiSeq2000 sequence platform. All analysis was performed using BMKCloud (www.biocloud.net, accessed on 3 May 2022). 

## 3. Results

### 3.1. Identification of BAHD Family Genes Welsh Onion

The BAHD superfamily is known to contain a conserved transferase domain in various plant species. We identified all the BAHD genes in the Welsh onion genome with the transferase domain (PF02458) and found a total of 97 genes. To confirm this identification, we analyzed all AfBAHDs with conserved domains using PfamScan, InterPro, and SMART tools. The results are listed in Appendix A, showing that all AfBAHDs possessed the transferase domain. The amino-acid count of AfBAHDs ranged from 89 to 761 (See Appendix A for additional information on AfBAHDs). Our results are consistent with the previous identification of the BAHD gene family in plant genomes, indicating the reliability of our analysis.

### 3.2. Location of BAHDs Family Genes in Welsh Onion Chromosomes

To examine the locations of AfBAHDs, we conducted a search for gene annotations and generated a location map (Figure 1). We found that AfBAHDs were unevenly distributed across all chromosomes of Welsh onion. AfBAHDs (97 in total) were located on Chr3 (27.8%), followed by Chr1 (20.6%), Chr6 (12.3%), Chr4 (11.3%), Chr2 (10.3%), Chr5 (7.2%), Chr7 (7.2%), and Chr8 (3.0%). To investigate the collinearity and duplication relationships of AfBAHDs in the onion genome, we analyzed them using MCScanX. We found that there was no collinearity or duplication among these AfBAHDs (results not shown). This suggests that AfBAHDs may have undergone specialization and differentiation during the process of evolution.

### 3.3. Phylogenetic Analysis of the BAHD Family Genes in Welsh Onion

The BAHD acyltransferase gene family comprises numerous members. To gain a better understanding of the function of AfBAHDs in different lineages, we constructed an evolutionary tree that incorporated other reported BAHD proteins from *Arabidopsis thaliana*, *Oryza sativa*, and *Zea mays*. As a result, all AfBAHDs were divided into seven clades on the basis of their amino-acid sequences (Figure 2). The largest clade, Clade Ia, consisted of 22 members, while Clade Ib contained 19 members, including AtDCR and LuDCR1. Clade IIa included 17 AfBAHD members, such as AtFACT, AtASFT, AtDCF, and StFHT. Five members of AfBAHDs were placed in Clade IIb, which also included OsAT4 and ZmPCAT. Clade III was the smallest with only two members. Clade IV contained 11 AfBAHDs, including HCBT, HQT, AsHHT, AtHCT, and NtHCT. Clade V consisted of three AfBAHDs and included AtCER26, AtCER26-LIKE, AtCER2, and ZmCER2/Glossy. By utilizing the identified AfBAHDs and previously reported BAHDs, we constructed an evolutionary tree, which classified these genes into seven branches, consistent with previous findings, further supporting the diversity of BAHD functions. Notably, three genes homologous to onion CER2 were categorized in Clade V. These three genes were named CER2-LIKE1, CER2-LIKE2, and CER2-LIKE3, and they play a crucial role in the synthesis of epidermal wax in Welsh onion. Their inclusion in the same clade with key genes involved in long-chain fatty acid metabolism indicates their potential involvement in the biosynthesis of lipid-based compounds in plants.

### 3.4. Gene Structures and Function Domains of BAHD Family Genes in Welsh Onion

To understand the phylogenetic relationships among AfBAHD members, we conducted an analysis of their complete mRNA and protein sequences. First, AfBAHDs were clustered as shown in Figure 3A. We then examined the motifs present in Welsh onion BAHDs and identified 10 conserved motifs using the MEME database. The result suggests that the majority of AfBAHDs contained Motif1 and Motif3, which are characterized by the HXXXD and DFGWG motifs, respectively. These motifs are specific to the transferase family. We also investigated the gene structures of AfBAHDs and found that 36 genes contained no introns, 39 genes contained one intron, 16 genes contained two introns, four genes contained three introns, and two genes had five introns. These results suggest that all AfBAHDs feature at least two transferase-related motifs, with both Motif1 and Motif3 present in all of them (see Figure 3B). All groups exhibited a similar motif composition, indicating that these genes possess general characteristics of the BAHD gene family and supporting the results of the evolutionary relationships.

### 3.5. Analysis of cis-Acting Element and Gene Expression in the BAHD Family Genes of Welsh Onion

To understand the regulation of AfBAHDs, we performed an analysis of the *cis*-acting elements in the promoters of 97 BAHDs in Welsh onion. We detected *cis*-acting elements related to biotic and abiotic responses, including MeJA responsiveness, defense and stress responsiveness, and drought, cold, ABA (abscisic acid), GA (gibberellin), auxin, and light responsiveness (Figure 4A). These results indicate that AfBAHDs are regulated by multiple signals and play a role in various biotic and abiotic stress responses. Our findings further support the role of BAHDs in plant adaptation to diverse external environments, highlighting their significance in plant growth and development. 

In order to further understand the roles of these genes in adapting to the environment, we analyzed their expression in leaves and stems using RNA-seq. The results showed that 17 AfBAHDs were upregulated (|log_2_FC| > 1 and FDR < 0.05) (Figure 4B). Further enrichment analysis and annotation of 97 AfBAHDs using KEGG showed that these genes were enriched in various secondary metabolic pathways, such as flavonoid, cutin, and wax biosynthesis (Figure 4C).

### 3.6. Sequence Analysis of CER2-LIKE Genes in Welsh Onion

The CER2-LIKEs play a crucial role in wax metabolism, and we identified three members in Clade V that share classification with other reported CER2-LIKEs in different species. Sequence analysis showed that AfCER2-LIKEs encoded sequences consisting of 419, 423, and 383 amino acids. Multiple alignment analysis demonstrated the conservation of the structures of AfCER2-LIKEs across species, with their encoded amino-acid sequences exhibiting high similarity to other CER2-LIKEs. We also found that the HXXXD motif was present in AfCER2-LIKEs (Figure 5). Previous studies have demonstrated that the HXXXD motif affects the function of CER2 [4,32]. In Welsh onions, the His residue in the HXXXD motif of AfCER2-LIKEs is replaced by Leu, suggesting that the function of AfCER2-LIKE3 may differ from that of AfCER2-LIKE1 and AfCER2-LIKE2. Further investigation of the functions of AfCER2-LIKEs could explain the differences in the HXXXD motif among CER2 homologs and enhance our understanding of the regulation of wax metabolism in Welsh onions.

### 3.7. 3D Modeling of AfCER2-LIKEs

To understand the protein differences among AfCER2-LIKEs, we used Phyre to model AfCER2-LIKE1, AfCER2-LIKE2, and AfCER2-LIKE3. The predicted results showed that all AfCER-LIKEs are transferases, and PDBTitle divided them into different types of transferases (Figure 6). These results further suggest that AfCER2-LIKEs not only have different functions, but may also have a wide range of substrate activities.

### 3.8. Expression of AfCER2-LIKE Genes in Wildtype Welsh Onion and Wax-Deficient Welsh Onion

To understand the role of CER2-LIKEs in wax metabolism, we analyzed the expression of CER2-LIKEs in different tissues of the wildtype BG and waxy mutant GLBG (Figure 7A). The expression of AfCER2-LIKEs was mainly observed in flower and scape regions. AfCER2-LIKE1 also exhibited expression in leaves, suggesting its involvement in wax synthesis in the epidermis of Welsh onion leaves (Figure 7B). Further analysis revealed that the expression of CER2-LIKE1 was almost undetectable in GLBG. In our previous research, we reported that GLBG is a mutant that lacks leaf cuticular wax [33]. These findings indicate the critical role of CER2-LIKE1 in the epidermal wax metabolism of Welsh onion leaves and suggest that the absence of its expression in any tissue of the mutant may be a key factor contributing to the waxy phenotype. 

### 3.9. Expression of AfCER2s Genes under Abiotic Stress

The expression profiles of AfCER2-LIKE genes were analyzed under salt stress, drought stress, high-temperature stress, and cold stress at different timepoints (0, 4, 8, 12, 24, 48, and 72 h). Heat stress upregulated the expression of AfCER2-LIKEs (Figure 8A). Drought stress resulted in irregular expression patterns of AfCER2-LIKE1 and AfCER2-LIKE2, while the expression of AfCER2-LIKE3 was reduced under drought conditions (Figure 8B). Salt stress inhibited AfCER2-LIKE expression. Low temperature, on the other hand, reduced the expression of AfCER2-LIKE1, while AfCER2-LIKE2 and AfCER2-LIKE3 showed an opposite expression trend under low-temperature stress (Figure 8D). These findings highlight the potential involvement of Clade V genes of the BAHD family, AfCER2-LIKEs, in Welsh onion wax according to evolutionary and sequence analyses. Furthermore, the differential expression patterns of AfCER2-LIKE1, AfCER2-LIKE2, and AfCER2-LIKE3 under different abiotic stresses suggest their nonredundant roles in wax metabolism in Welsh onion. These results provide valuable insights for further studying the regulation of wax metabolism in Welsh onions.

## 4. Discussion

Enzymes involved in plant acetylation modification have been extensively reported [1,5,14]. Genomic studies have revealed the widespread presence of acyl coenzymes in land plants, demonstrating the important role of acylation modification in primary and secondary metabolism of plants, ultimately influencing plant growth and development [1]. Among the BAHDs family, CER2-LIKEs have been shown to be involved in the synthesis of VLCFA [12,13,14]. Welsh onion exhibits a distinct wax texture on its leaf epidermis [33]. While the BAHD family has been studied in model plants such as *Arabidopsis* and rice, research on wax synthesis in Welsh onion has been limited due its large genome size. However, the release of the Welsh onion genome has significantly facilitated the study of wax metabolism in this species [34]. 

In this study, we performed a comprehensive genome-wide analysis of the BAHD family in Welsh onion. A total of 97 BAHD family genes were identified. The AfBAHDs were categorized into seven clades on the basis of their phylogenetic relationship. Compared to the number of BAHD genes in other species, such as *Musa acuminate* [7], *Populus* [35], *Oryza sativa* [36], and *Arabidopsis thaliana* [1], the number of BAHD genes in Welsh onion is much higher. This may be due to the large genome size of Welsh onion. Compared to species with a smaller BAHD gene family, the number of BAHDs in Welsh onion is almost twice that of these species, indicating a significant expansion of the BAHD gene family in onions. In the common ancestor of land plants and algae, only a limited number BAHDs (about 1–5) were present. BAHDs rapidly expanded in land plant species, leading to metabolic diversity in plants. Compared to other species with convergent BAHDs, the expansion of BAHDs in onions provides an enzymatic basis for the diverse metabolic pathways in Welsh onion and for their adaptation to various environments. In addition, both structural domain analysis and motif analysis indicated that AfBAHDs possess the unique characteristics of this family, consistent with previous reports [7,35]. 

Exon/intron composition analysis showed that the number of introns in AfBAHDs varied between zero and four, and most AfBAHDs contained either one intron or no introns at all. This observation is consistent with previous findings in other species. We also analyzed the upstream regulatory sequence of AfBAHDs. The results demonstrated the presence of *cis*-acting elements associated with biotic and abiotic stress, including pathogen defense, damage, drought, cold, ABA, MeJA, GA, auxin, and light responsiveness. Therefore, these findings further support the critical role of the BAHD family in plant adaptation to terrestrial environments, response to abiotic stress, and interactions with biotic factors.

Wax serves crucial functions in plant stress resistance and pollen development [1,9]. VLCFAs and their derivatives are the main components of plant epidermal wax [16,37]. The CER2-LIKE subfamily plays an important role in VLCFA metabolism, particularly the elongation of fatty acids through interaction with KCS6 (*β*-ketoacyl-coenzyme A synthase) [11,13,18]. Here, we identified the members of the CER2-LIKE subfamily of the BAHD family in Welsh onion. Further sequence analysis revealed the presence of the HXXXD motif in AfCER2-LIKEs in Welsh onion. We analyzed the function and expression of *CER2-LIKEs* in BG and GLBG. The results showed that *AfCER2-LIKE1* was expressed in the BG leaves but showed minimal expression in the GLBG, suggesting that *AfCER2-LIKE1* plays a critical role in the synthesis of leaf epidermal wax. The high expression levels of *AfCER2-LIKE2* and *AfCER2-LIKE3* in flowers suggest their involvement in pollen lipid development. In addition, we assessed the expression profiles of *AfCER2-LIKEs* under different stress conditions. Drought stress led to irregular expression of *AfCER2-LIKE1* and *AfCER2-LIKE2*, while it reduced the expression of *AfCER2-LIKE3*. Under high-temperature stress, the expression levels of AfCER2-LIKEs were upregulated, suggesting that AfCER2-LIKEs, key genes involved in high-temperature-induced wax metabolism in Welsh onion, increase the wax load on the epidermis of Welsh onion and enhance their heat tolerance. While low temperature reduced the expression of *AfCER2-LIKE1*, *AfCER2-LIKE2* and *AfCER2-LIKE3* showed the opposite trend. These findings suggest that the expression of AfCER2-LIKEs is regulated by environmental conditions, and these genes likely play a crucial role in plant adaptation to the environment.

## 5. Conclusions

In this study, we characterized BAHD genes in Welsh onion. A total of 97 BAHD genes were identified in Welsh onion genome. We conducted a comprehensive analysis to probe various aspect of these genes, including their structural features, conserved motifs, phylogenetic relationships, and upstream regulatory sequences. Furthermore, we characterized the expression patterns of AfCER2-LIKEs under various stress conditions. Our findings revealed that the AfCER2-LIKEs, members of the BAHD gene family, may play crucial roles in wax metabolism and in abiotic stress response. These results contribute to our understanding of the molecular mechanisms underlying wax metabolism and the response to abiotic stress in Welsh onion, highlighting the importance of the BAHD gene family, particularly AfCER2-LIKEs, in these processes.

## Figures and Tables

**Figure 1 genes-14-01286-f001:**
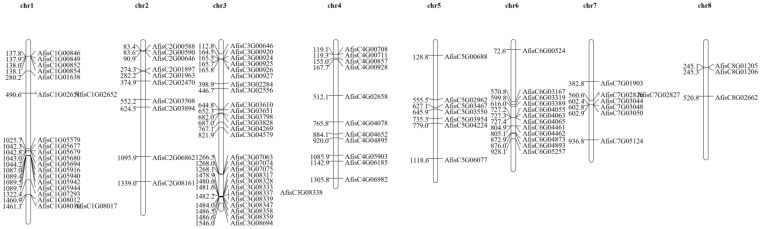
The Location of AfBAHDs in Welsh onion chromosomes. The distribution of the AfBAHDs gene on eight chromosomes. The number of chromosomes is displayed at the top of each chromosome. The AfBAHDs gene name is displayed on the left side of each chromosome. The scale of the genome size is given on the left.

**Figure 2 genes-14-01286-f002:**
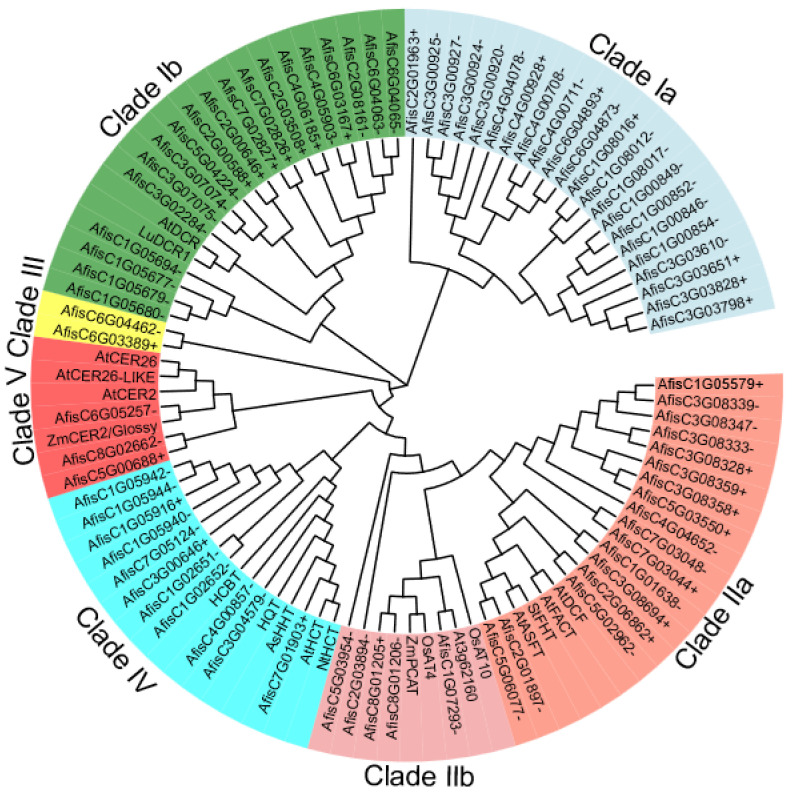
Phylogenetic relationships of AfBAHD proteins. A phylogenetic tree was constructed in MEGA6.0 on the basis of the full-length amino-acid sequences of previously identified BAHD proteins, with the accession numbers of sequences used in the alignment provided in the methods. ClustalW was used with 10,000 bootstrap replicates, and the tree was built using the neighbor-joining (NJ) method. All BAHDs belong to seven different clades; different clade members are labeled different colors.

**Figure 3 genes-14-01286-f003:**
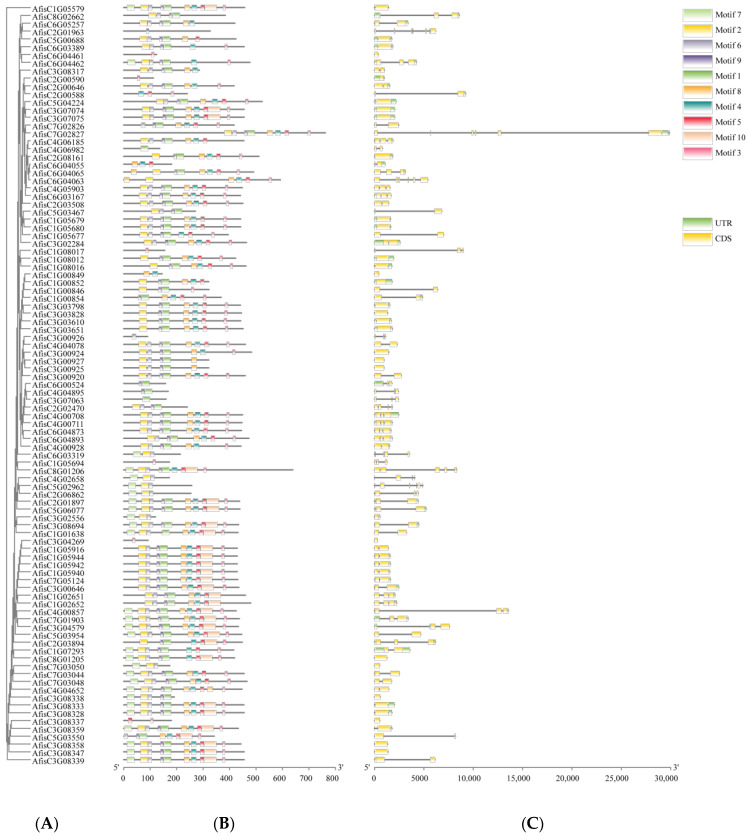
Gene structure and multiple motifs analysis of BAHD family in Welsh onion. (**A**) The evolutionary relationship analyses of BAHDs based on phylogenetic relationships. (**B**) The motif analyses of AfBAHDs. Conservation motifs of BAHDs were analyzed using the MEME online tool, by setting the upper limit at 10 for identifying conservation motifs. Each box represents a conserved motif, as shown in the legend on the right. (**C**) The exon–intron analyses of AfBAHDs. Exon and intron information was extracted from the genome annotation file. The green box represents noncoding regions, and the yellow box represents coding regions. Legends are shown on the right.

**Figure 4 genes-14-01286-f004:**
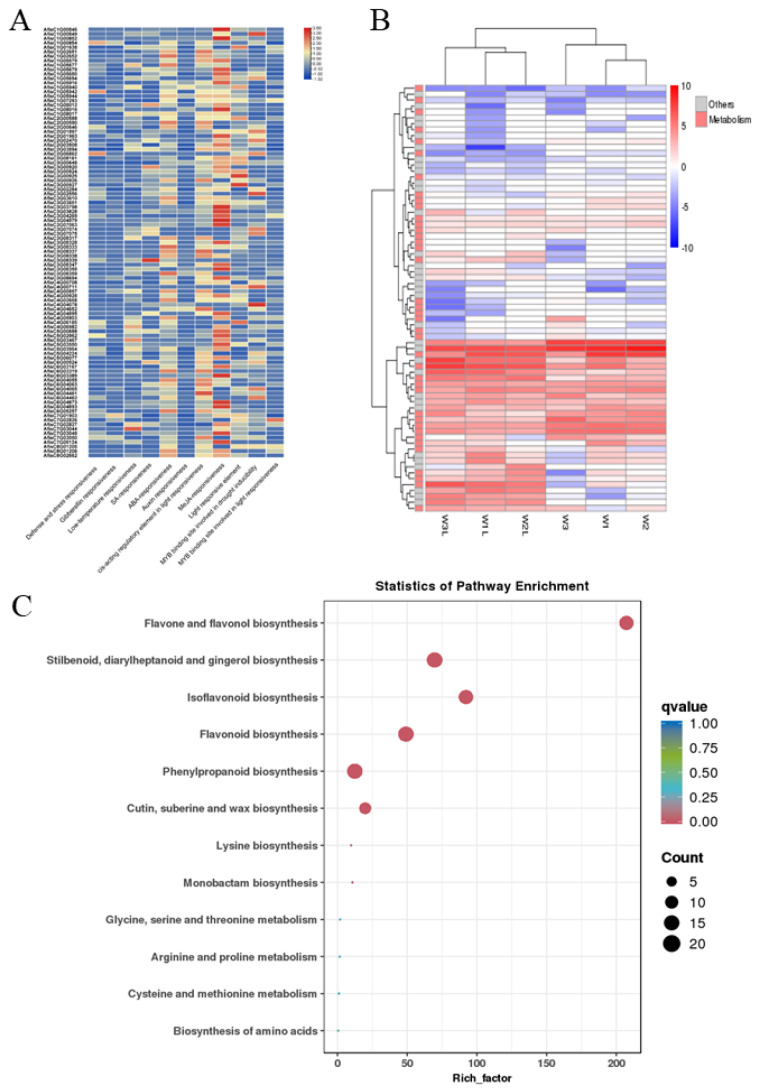
Analysis of *cis*-regulatory elements and expression of the AfBAHD genes. (**A**) The number of *cis*-regulatory elements of AfBAHDs. (**B**) The gene expression of AfBAHDs in Welsh onion leaves and scape. W1L, W2L, and W3L denote Welsh onion leaves. W1, W2, and W3 denote Welsh onion scape. (**C**) KEGG enrichment analysis of AfBAHDs. The analyses were performed using BMKCloud (www.biocloud.net, accessed on 3 May 2022).

**Figure 5 genes-14-01286-f005:**
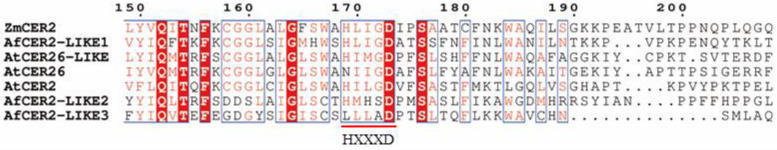
Sequence analysis of CER2-LIKEs in Welsh onion. Using DNAMAN, a multiple sequence alignment of AfCER2-LIKEs was performed with the homologous genes of CER2 previously reported. The motif of HXXXD is marked by a red line.

**Figure 6 genes-14-01286-f006:**

The 3D models of AfCER-LIKEs constructed using the online tools of Phyre. (**A**) The 3D model of AfCER2-LIKE1. (**B**) The 3D model of AfCER2-LIKE2. (**C**) The 3D model of AfCER2-LIKE3.

**Figure 7 genes-14-01286-f007:**
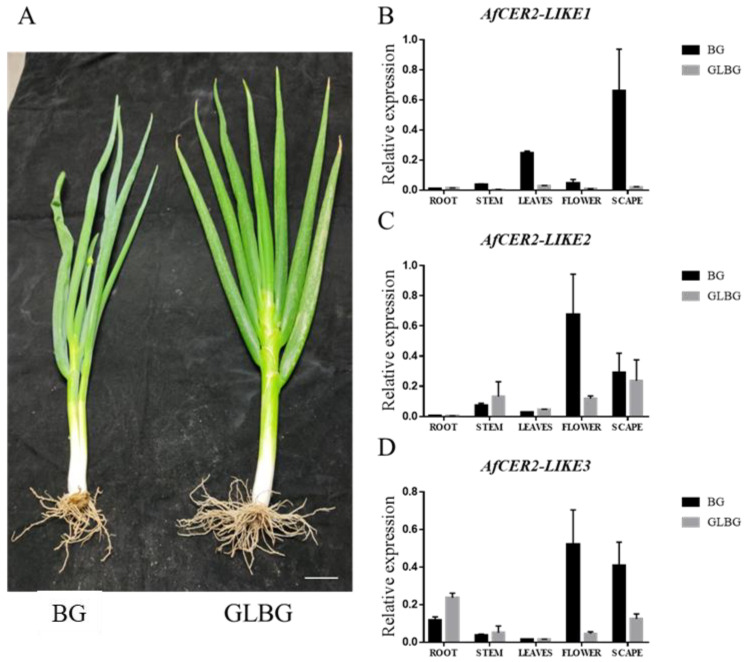
Relative expression of AfCER2-LIKEs in different tissue. Analysis of AfCER2 expression in the roots, stems, leaves, flowers, and scape of Welsh onion using quantitative real-time PCR. (**A**) Wildtype Welsh onion “BG” and wax-deficient mutant Welsh onion “GLBG”. (**B**) Expression of AfCER2-LIKE1 in different tissues of Welsh onion. (**C**) Expression of AfCER2-LIKE2 in different tissues of Welsh onion. (**D**) Expression of AfCER2-LIKE3 in different tissues of Welsh onion. Each experiment was performed with three biological replicates.

**Figure 8 genes-14-01286-f008:**
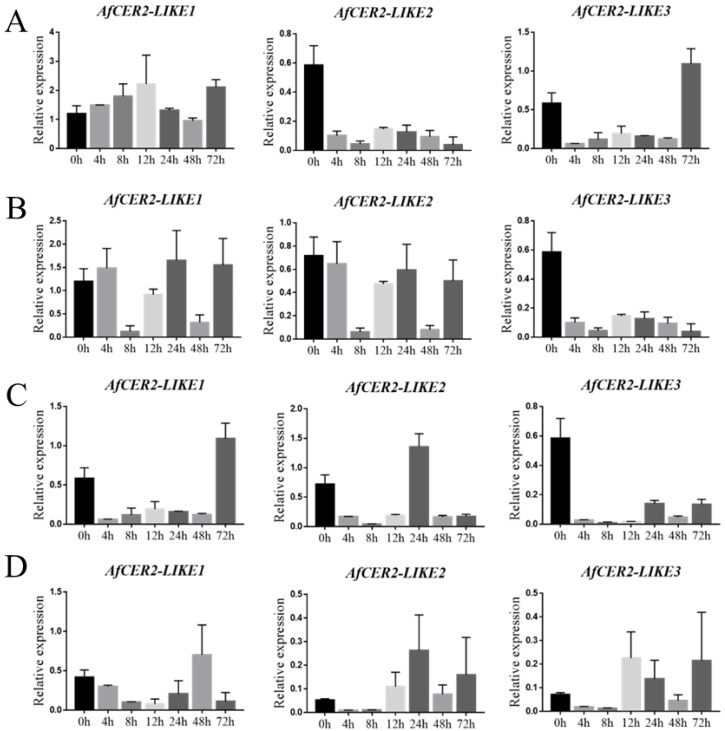
The expression of AfCER2-LIKEs under different stress. (**A**) CER2-LIKEs of Welsh onion under heat stress. (**B**) CER2-LIKEs of Welsh onion under drought stress. (**C**) CER2-LIKEs of Welsh onion under salt stress. (**D**) CER2-LIKEs of Welsh onion under cold stress. Each experiment was performed with three biological replicates.

## Data Availability

Not applicable.

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
