# Peer review of "Genome-Wide Analysis of the BAHD Family in Welsh Onion and CER2-LIKEs Involved in Wax Metabolism"

_genes, 2023, doi:10.3390/genes14061286_

Round 1
Reviewer 1 Report
The manuscript “Genome-Wide Analysis of BAHDs Family in Allium fistulosum
and Expression of CER2-LIKEs under Abiotic Stress” described the study of gene structures, cis-acting elements, and phylogenetic analysis of BAHD proteins in different plant species and its homologs CER2-LIKEs genes in Welsh onion. Tissue specific expression and abiotic stress treatment expression analysis of CER2-LIKEs genes were also performed in this study.
Major concerns
-
Did the authors perform any biotic stress experiments since some of the cis-acting elements in the promoter of BAHD genes were involved in regulating the pathogen defense mechanism?
-
The process of wax synthesis in leaves may vary during different stages of plant development. Which stages of plant tissue of Welsh onion were used for expression analysis?
-
CER2-LIKEs are important for wax metabolism. Did the biochemical GC-MS analysis that focus on intracellular fatty acid/cellular lipidome was done in this study?
-
Did the authors perform any over-expression or loss of function approach of the CER2-LIKEs genes to further understand the functionality of CER2-LIKEs gene in the Welsh onion since its genome is available?
Minor errors
Page 1, Line 17, Very Long Chain Fatty Acids (VLCFA), please indicate the full name
Page 2, Line 46, Aliphatic Suberin Feruloyl Transferase (ASFT), please be specific
Page 3, Line 116, were listed in Table S1, grammar error
Page 3, Line 123, were unevenly distributed, grammar error
Page 4, Line 131, Arabidopsis thaliana, Oryza sativa and Zea mays should be italicized
Page 4, Line 154, in Welsh onion were analyzed, grammar error
Page 8, Line 206, Musa acuminate, Oryza sativa and Arabidopsis thaliana should be italicized
Page 8, Line 222, CER2-Likes, put a dash in between
Page 8, Figure Legends (b). CER2-LIKEs of Welsh onion under cold stress should be (d), wrong labeling, the labeling of relative expression of AfCER2-LIKE1, AfCER2-LIKE2, and AfCER2-LIKE3 in the figure was confusing
Author Response
Thank you for your comments and professional opinions on our manuscript. These suggestions have improved the quality and academic rigor of our manuscript. Following your advice, we have made some revisions in the revised version. Additionally, the manuscript has been language edited. Furthermore, we would like to provide some details below.
Major concerns
1.Did the authors perform any biotic stress experiments since some of the cis-acting elements in the promoter of BAHD genes were involved in regulating the pathogen defense mechanism?
Response:
We did not conduct any biotic stress experiments. Through analysis of the BAHD gene promoter, we did identify some elements that regulate pathogen defense mechanisms, but we were more focused on identifying genes in BAHD that might be involved in wax metabolism. Conducting some biotic stress experiments is a good suggestion.
2.The process of wax synthesis in leaves may vary during different stages of plant development. Which stages of plant tissue of Welsh onion were used for expression analysis?
Response:
We agree that wax synthesis in leaves may differ at different stages of plant development. We have analyzed the expression of two-month-old Welsh onion.
3.CER2-LIKEs are important for wax metabolism. Did the biochemical GC-MS analysis that focus on intracellular fatty acid/cellular lipidome was done in this study?
Response:
Analyzing the fatty acid/cell lipid composition inside cells is a good method to study overall plant lipid metabolism. However, our work is mainly focused on onion epidermal wax, which is important for onion's resistance against biotic stress.
4.Did the authors perform any over-expression or loss of function approach of the CER2-LIKEs genes to further understand the functionality of CER2-LIKEs gene in the Welsh onion since its genome is available?
Response:
To further understand the function of CER2-LIKEs in Welsh onion, we analyzed the expression of a wax-deficient mutant in our previous work. While the genome is available, the cost of gene transformation is too high, and due to the large size of the onion genome, there are currently no reports of genetic transformation in onion. We have attempted genetic transformation but have been unable to obtain transgenic lines.
Minor errors
Page 1, Line 17, Very Long Chain Fatty Acids (VLCFA), please indicate the full name
Response:
We have carefully checked that the full name of VLCFA is Very Long Chain Fatty Acids.
Page 2, Line 46, Aliphatic Suberin Feruloyl Transferase (ASFT), please be specific.
Response:
Thanks for the reviewer's reminder. We have provided additional explanations at this point.
Page 3, Line 116, were listed in Table S1, grammar error
Response:
We have made corrections to the grammar error here and have further polished the language throughout the manuscript.
Page 3, Line 123, were unevenly distributed, grammar error
Response:
Thanks for the reviewer's reminder. We have made corrections to the grammar error here and have further polished the language throughout the manuscript.
Page 4, Line 131, Arabidopsis thaliana, Oryza sativa and Zea mays should be italicized
Response:
We have italicized the Latin names of species throughout the entire manuscript.
Page 4, Line 154, in Welsh onion were analyzed, grammar error
Response:
Thanks for the reviewer's reminder. We have corrected the grammar error here.
Page 8, Line 206, Musa acuminate, Oryza sativa and Arabidopsis thaliana should be italicized
Response:
We have italicized the Latin names of species throughout the entire manuscript.
Page 8, Line 222, CER2-Likes, put a dash in between
Response:
We have added a dash here.
Page 8, Figure Legends (b). CER2-LIKEs of Welsh onion under cold stress should be (d), wrong labeling, the labeling of relative expression of AfCER2-LIKE1, AfCER2-LIKE2, and AfCER2-LIKE3 in the figure was confusing
Response:
Thanks for the reviewer's reminder. We have made modifications to all the images in the manuscript.
Reviewer 2 Report
Authors haves analysed BAHDs family in Allium fistulosum, however I found the analysis they perform is not enought. The identification of candidate genes with Transferase domain is fine as a first step however iterative search steps is always needed to be sure all possible copies (included pseudogenes) are identified. Moreover, the location of genes in chromosomes alone have no relevance as the genome is already available (data used for identification of candidate genes) and no further analysis is done (as it can be evolutive relationships). In this part, also methods are not described sufficiently, as there are not information about threshold to consider hit as candidate gene or not, for instance.
Then, authors work with CER2-LIKE genes and directly assume the same function with any biochemical support. In this kind of enzymes small changes in sequence can affect specially substrate recognition, not even a simulation with AtCER2 protein structure is performed where comparison of significant aminoacids can be confronted.
Only a few qRT-PCR are performed in limited conditions and some functional relation is presented with no more evidence than weak correlations. Without biochemical confirmation and with weak correlations (only clear in a few cases) functional conclusions are mostly expeculative.
Authors need to do more extensive analysis of this gene family, increasing phylogenetic analysis with many more members of other species, as BAHD genes have evolved in different species to display different functions (recognition of different substrate is a different function). Moreover, if bioinformatics methods are the main objecive, is even more important to use any available resource as CER2 3D protein structure and using it for modelling. And in any case, without biochemical confirmation or functional genetics analysis (what is not so difficult) any functional conclusion is mainly expeculative.
It has been a difficult reading, gramatically and specially verbs can be clearly improved and I suggest authors to use professional services before sending manuscript.
Author Response
Thank you for your comments and professional opinions on our manuscript. These suggestions have improved the quality and academic rigor of our manuscript. Following your advice, we have made some revisions in the revised version. Additionally, the manuscript has been language edited. Furthermore, we would like to provide some details below.
Major concerns
Authors haves analysed BAHDs family in Allium fistulosum, however I found the analysis they perform is not enought. The identification of candidate genes with Transferase domain is fine as a first step however iterative search steps is always needed to be sure all possible copies (included pseudogenes) are identified. Moreover, the location of genes in chromosomes alone have no relevance as the genome is already available (data used for identification of candidate genes) and no further analysis is done (as it can be evolutive relationships). In this part, also methods are not described sufficiently, as there are not information about threshold to consider hit as candidate gene or not, for instance.
Then, authors work with CER2-LIKE genes and directly assume the same function with any biochemical support. In this kind of enzymes small changes in sequence can affect specially substrate recognition, not even a simulation with AtCER2 protein structure is performed where comparison of significant aminoacids can be confronted.
Only a few qRT-PCR are performed in limited conditions and some functional relation is presented with no more evidence than weak correlations. Without biochemical confirmation and with weak correlations (only clear in a few cases) functional conclusions are mostly expeculative.
Authors need to do more extensive analysis of this gene family, increasing phylogenetic analysis with many more members of other species, as BAHD genes have evolved in different species to display different functions (recognition of different substrate is a different function). Moreover, if bioinformatics methods are the main objecive, is even more important to use any available resource as CER2 3D protein structure and using it for modelling. And in any case, without biochemical confirmation or functional genetics analysis (what is not so difficult) any functional conclusion is mainly expeculative.
Response:
- In fact, we conducted further searching based on the HMM generated by AfBAHDs to identify all possible copies. However, we did not describe this in the manuscript. Considering the rigor of the manuscript, we have added this information in the method section.
- Initially, we conducted further analysis of chromosome location. Interestingly, we found that there were almost no duplications and collinearity among these genes. Previous studies have not conducted further analysis of the chromosomal location of BAHD gene family members. Considering the size of this extensive gene family and its diverse functions, this may be related to the evolutionary relationships you mentioned. We believe that the location of BADHs helps explain how BAHDs produce a wide range of function.
- We did not provide the threshold for the selection during BAHDs identification. However, we controlled it during this step. Furthermore, we conducted further screening based on the conservative structure domain of BAHDs. These details are pointed out in the revised manuscript.
- The simulation using the AtCER2 protein structure is a good method for discovering significant amino acids. However, the function of CER2-LIKEs is still controversial, and their function may depend on acyltransferase characteristics in different species. What is more, we lack knowledge about molecular docking in simulation.
- To be frank, our previous manuscript did not have enough information about the function of AfCER2-LIKEs. In the new manuscript, we have supplemented the expression analysis of AfCER2-LIKE genes in the mutant of Welsh onion. These results support that AfCER2-LIKE1 involved in metabolism of wax in leaf epidermis.
- We agree with your view only considering gene family analysis. It is necessary to increase the systematic phylogenetic analysis with many more members of other species. However, the BAHD gene family is extensive and has various functions. We considered that adding more unknown functional BAHDs would make our understanding of the function more confusing, and the figures would become more complex. Besides, we specifically want to highlight AfCER2-LIKEs among the BAHDs. In the beginning, we attempted to establish an evolutionary tree of all the BAHD gene functions in Arabidopsis, rice, and tobacco. However, the resulting tree was too large to observe some of the information we needed. As a result, we only retain the BAHDs with confirmed functions.
- Biochemical and functional genetic analysis is a good way to identify AfCER2-LIKEs. In the new version, we provide 3D model analysis of AfCER2-LIKEs. At the same time, we have supplemented the expression data of AfCER2-LIKEs in a Welsh onion mutant. These results support the participation of AfCER2-LIKE1 in the cuticular wax metabolism of Welsh onion leaves. This mutant was reported to be deficient in wax..
Identification and Characterization of a Glossy Mutant in Welsh Onion (Allium fistulosum L.). Sci. Hortic. 2017, 225, 122–127, doi: 10.1016/j.scienta.2017.05.014.
Minor errors
It has been a difficult reading, gramatically and specially verbs can be clearly improved and I suggest authors to use professional services before sending manuscript.
Response:
Thanks for the reviewer’s suggestions, the manuscript has been polished.
Reviewer 3 Report
In this manuscript, the authors investigated the BAHD family in the genome of Welsh onion. The authors studied this gene family via phylogenetic analysis, gene structure and motif analyses. Further transcriptional analysis were carried out to examine the expression pattern of these genes in response to abiotic stress through qPCR. However, more experimental evidence should be provided to support the potential roles of CER2 as the authors proposed. My comments are listed in detail as follows.
1. In Plant Materials, more detailed information of Welsh onion age and conditions should be provided.
2. In the Materials and Methods, what is the length of promoters used for cis-acting element analysis?
3. For qPCR, what is the reference gene and primer information
4. Figure legends should be provided for all figures.
5. For each result section, a brief conclusion should be provide after result analysis.
6. The discussion section should provide more informative conclusions and discussion.
7. More experimental evidence should be provided to support their roles in response to abiotic stress.
Author Response
Thank you for your comments and professional opinions on our manuscript. These suggestions have improved the quality and academic rigor of our manuscript. Following your advice, we have made some revisions in the revised version. Additionally, the manuscript has been language edited. Furthermore, we would like to provide some details below.
In this manuscript, the authors investigated the BAHD family in the genome of Welsh onion. The authors studied this gene family via phylogenetic analysis, gene structure and motif analyses. Further transcriptional analysis were carried out to examine the expression pattern of these genes in response to abiotic stress through qPCR. However, more experimental evidence should be provided to support the potential roles of CER2 as the authors proposed. My comments are listed in detail as follows.
- In Plant Materials, more detailed information of Welsh onion age and conditions should be provided.
Response:
Thanks for the reviewer’s reminder, we have added this information to the methods section.
- In the Materials and Methods, what is the length of promoters used for cis-acting element analysis?
Response:
The promoter length used for cis-regulatory element analysis is 2,000bp, and we have added these details to the methods section.
- For qPCR, what is the reference gene and primer information
Response:
Thanks for the reviewer’s reminder, we have added the ID and primer information for the reference gene used in qPCR analysis.
- Figure legends should be provided for all figures.
Response:
We have made modifications to all images in the manuscript and have provided a legend for each of them.
- For each result section, a brief conclusion should be provide after result analysis.
Response:
We have added analysis and brief conclusions for each of the results presented.
- The discussion section should provide more informative conclusions and discussion.
Response:
Thanks for the reviewer’s reminder, we have addressed this suggestion by adding supplementary information in the discussion section of the manuscript.
- More experimental evidence should be provided to support their roles in response to abiotic stress.
Response:
Our study focuses on the stress response of the CER2 branch in the BAHDs. We have added information on the expression of CER2-LIKE genes in wax mutants to confirm their crucial role in wax metabolism and provided data on their expression under abiotic stress. We are particularly interested in understanding the role of wax metabolism in alleviating abiotic stress in Allium fistulosum.
Round 2
Reviewer 1 Report
All my concerns have been sufficiently addressed in the revision.
Author Response
All my concerns have been sufficiently addressed in the revision.
Response:
Thanks again for the reviewer's comments and professional opinions. These suggestions have improved the quality and academic rigor of our manuscript.
Reviewer 2 Report
I thank authors for their reply and the comments to the different points I did highlight. Authors have improved the manuscript with 3D structure prediction which are interesting. However, I have several concerns and I think that some points must be addressed.
My main concern is about cis-elements analysis. To give any conclusion just with identification of consense motifs using Plant-CARE is clearly too especulative. Appart of the limited significance of finding consense small sequences, there is no comparison with what can be detected in other sequences with no similar functionality at all. What will be detected using any random sequence in the genome? What will be detected using promoters of any other gene family not related with biotic or abiotic stress? Are some of the motifs over-represented? This kind of questions must be addressed because if you can find a similar result using any other set of sequences, the result has no meaning at all without experimental support in vivo.
I have also some suggestions for authors in order to improve the manuscript and generate mor impact:
a) The prediction of 3D structure is interesting, and Phyre is ok, however I think it would be worthy if authors also check if the EMBL have the prediction from AlphaFold2 (I have checked they have 133 predictions of Allium fistulosum proteins). If authors could have 2 predictions and compare them the discussion will be better supported.
b) Number or distribution of genes by chromosomes has no direct relation with gene function so it cannot be related with the variety of functions in BAHD family. In any case, having duplicates is a "source" for diversification but anything else.
c) Tha majority means >50%, other way to describe is needed, like “The chromosome with a higher number of BAHDs (27) is chromosome 3 (27.8%), followed by …”
d) Description of introns pattern in BAHD genes is confusing:
“…39 genes contained one intron, 36 genes contained no introns, and 2 genes contained up to 5 introns. Among the genes with introns, 14 had 2 introns, and 4 had 3 introns.”
With this sentence it s not clear which are the genes with 2-3 introns, Because is indicated that 39 has one, 36 do not have, and 2 have up to 5 (so more than one and less than 6)…but then 14 has 2 introns a 4 have 3… but only 2 had up to 5… I think they must be in the same sentence to be clear:
“…36 genes contained no introns, 39 genes had only one intron, 14 had 2 introns, 4 had 3 introns and 2 genes contained 5 introns.” However check numbers, as the total is 95 not 97.
e) Sinteny analysis between chromosomes it is easy to do and I think would be interesting to reflect duplication history.
f) In figure 6 the description os not correct. Is described A,B and C (that actually correspond with panels B, C and D, and there is no information fo what is panel A.
g) Authors have comented in their reply they have included expression analysis in a Welsh Onion mutant, however I see exactly the same figure and no reference to the mutant is present in the text.
Author Response
Thanks again for the reviewer's comments and professional opinions. These suggestions have improved the quality and academic rigor of our manuscript. Following your advice, we have made some revisions in the revised version and provide some details below.
Major concerns
I thank authors for their reply and the comments to the different points I did highlight. Authors have improved the manuscript with 3D structure prediction which are interesting. However, I have several concerns and I think that some points must be addressed.
My main concern is about cis-elements analysis. To give any conclusion just with identification of consense motifs using Plant-CARE is clearly too especulative. Appart of the limited significance of finding consense small sequences, there is no comparison with what can be detected in other sequences with no similar functionality at all. What will be detected using any random sequence in the genome? What will be detected using promoters of any other gene family not related with biotic or abiotic stress? Are some of the motifs over-represented? This kind of questions must be addressed because if you can find a similar result using any other set of sequences, the result has no meaning at all without experimental support in vivo.
Response:
We fully agree with your views on the analysis of cis-acting elements. Without any data to support such analyses, the results will be meaningless. Therefore, we analyzed the transcriptome data generated from previous studies and further analyzed the expression of AfBAHDs that we identified. In the short term, we cannot generate more data to support the response of AfBAHDs to various biotic or abiotic stressors. We analyzed AfBAHDs in leaves and pseudo-stems, these results support our view that AfBAHDs respond to environment condition. We have also added this section to our manuscript.
a) The prediction of 3D structure is interesting, and Phyre is ok, however I think it would be worthy if authors also check if the EMBL have the prediction from AlphaFold2 (I have checked they have 133 predictions of Allium fistulosum proteins). If authors could have 2 predictions and compare them the discussion will be better supported.
Response:
Thanks for the reviewer's suggestion. Using AlphaFold2 is indeed great and may be very useful for our work. We tried to search for the protein of interest on EMBL, but we couldn't find it. In fact, we wanted to use AlphaFold2 to predict these proteins, but we found that our region seems to have no access to it. This has made it impossible for us to continue this part of the work.
b) Number or distribution of genes by chromosomes has no direct relation with gene function so it cannot be related with the variety of functions in BAHD family. In any case, having duplicates is a "source" for diversification but anything else.
Response:
According to the reviewer’s comments, we found that there were indeed some issues in this part. In fact, we initially wanted to investigate the duplication status of these genes in genome. We found that there was no duplication or collinearity, which suggests that AfBAHDs may have become specialized and differentiated during evolution. This may explain the functional diversity observed in the AfBAHDs gene family.
c) Tha majority means >50%, other way to describe is needed, like “The chromosome with a higher number of BAHDs (27) is chromosome 3 (27.8%), followed by …”
Response:
Thanks for the reviewer’s reminder. We have revised the description.
d) Description of introns pattern in BAHD genes is confusing:
“…39 genes contained one intron, 36 genes contained no introns, and 2 genes contained up to 5 introns. Among the genes with introns, 14 had 2 introns, and 4 had 3 introns.”
With this sentence it s not clear which are the genes with 2-3 introns, Because is indicated that 39 has one, 36 do not have, and 2 have up to 5 (so more than one and less than 6)…but then 14 has 2 introns a 4 have 3… but only 2 had up to 5… I think they must be in the same sentence to be clear:
“…36 genes contained no introns, 39 genes had only one intron, 14 had 2 introns, 4 had 3 introns and 2 genes contained 5 introns.” However check numbers, as the total is 95 not 97.
Response:
Thanks for the reviewer’s suggestion. We have revised these descriptions and checked the number of genes and the number of introns accordingly.
e) Sinteny analysis between chromosomes it is easy to do and I think would be interesting to reflect duplication history.
Response:
We have added a description of this section in the manuscript and explained why there were no events of gene replication.
f) In figure 6 the description os not correct. Is described A,B and C (that actually correspond with panels B, C and D, and there is no information fo what is panel A.
Response:
Thank you to the reviewer for pointing out the issues with the manuscript. In the new version, we have check images and their descriptions.
g) Authors have comented in their reply they have included expression analysis in a Welsh Onion mutant, however I see exactly the same figure and no reference to the mutant is present in the text.
Response:
Thanks for the reviewer’s suggestions. We provided a photo of the mutant phenotype in Figure 6, analyzed gene expression in the mutant, and added a reference to the mutant in our manuscript.
Reviewer 3 Report
no further comment
Author Response
no further comment
Response:
Thanks again for the reviewer's comments and professional opinions. These suggestions have improved the quality and academic rigor of our manuscript.
Round 3
Reviewer 2 Report
I thank authors for their replies and comments, as all my comments have been addressed. I have only a minor comment at that point, about the comments regarding the distribution of BAHDs genes in chromosomes. Authors found no colinearity and they cannot associate clearly duplication events, that is ok with evolutive differentiation in activities, however is not well expressed when they say (remain only few copies). Actually there are 97 genes, and they probably are related, I don't think 97 are just a few. I think the text will be better if authors eliminate that sentence, or explain better what I think they want to say: that only in a few cases, probably the more recent duplications, the relationship can be clearly stablished.